# mTOR Driven Gene Transcription Is Required for Cholesterol Production in Neurons of the Developing Cerebral Cortex

**DOI:** 10.3390/ijms22116034

**Published:** 2021-06-03

**Authors:** Martin Schüle, Tamer Butto, Sri Dewi, Laura Schlichtholz, Susanne Strand, Susanne Gerber, Kristina Endres, Susann Schweiger, Jennifer Winter

**Affiliations:** 1Institute of Human Genetics, University Medical Center Mainz Johannes Gutenberg-University, 55131 Mainz, Germany; mschuele@students.uni-mainz.de (M.S.); T.Butto@imb-mainz.de (T.B.); Dewi.Hartwich@unimedizin-mainz.de (S.D.); lschlich@uni-mainz.de (L.S.); sugerber@uni-mainz.de (S.G.); susann.schweiger@unimedizin-mainz.de (S.S.); 2Focus Program of Translational Neurosciences, University Medical Center Mainz, 55131 Mainz, Germany; 3First Department of Internal Medicine, University Medical Center Mainz, 55131 Mainz, Germany; sstrand@uni-mainz.de; 4Department of Psychiatry and Psychotherapy, University Medical Center Johannes Gutenberg-University, 55131 Mainz, Germany; Kristina.Endres@unimedizin-mainz.de; 5German Resilience Centre, University Medical Center Mainz, 55131 Mainz, Germany

**Keywords:** mTOR, mTORC1, cholesterol, neurogenesis, SREBP, SP1, NF-Y

## Abstract

Dysregulated mammalian target of rapamycin (mTOR) activity is associated with various neurodevelopmental disorders ranging from idiopathic autism spectrum disorders (ASD) to syndromes caused by single gene defects. This suggests that maintaining mTOR activity levels in a physiological range is essential for brain development and functioning. Upon activation, mTOR regulates a variety of cellular processes such as cell growth, autophagy, and metabolism. On a molecular level, however, the consequences of mTOR activation in the brain are not well understood. Low levels of cholesterol are associated with a wide variety of neurodevelopmental disorders. We here describe numerous genes of the sterol/cholesterol biosynthesis pathway to be transcriptionally regulated by mTOR complex 1 (mTORC1) signaling in vitro in primary neurons and in vivo in the developing cerebral cortex of the mouse. We find that these genes are shared targets of the transcription factors SREBP, SP1, and NF-Y. Prenatal as well as postnatal mTORC1 inhibition downregulated expression of these genes which directly translated into reduced cholesterol levels, pointing towards a substantial metabolic function of the mTORC1 signaling cascade. Altogether, our results indicate that mTORC1 is an essential transcriptional regulator of the expression of sterol/cholesterol biosynthesis genes in the developing brain. Altered expression of these genes may be an important factor contributing to the pathogenesis of neurodevelopmental disorders associated with dysregulated mTOR signaling.

## 1. Introduction

Although changes in the activity of the mTOR pathway are associated with numerous neurodevelopmental disorders, the molecular basis of disease development remains poorly understood. The mTOR pathway is a central cellular signaling pathway that controls gene expression at multiple levels. While mTOR is well known for its role in regulating the translation of specific target mRNAs, it also governs RNA stability and gene transcription [1,2,3,4,5,6,7]. The evolutionarily conserved mTOR kinase comprises the core of two protein complexes, mTORC1 and mTOR complex 2 (mTORC2) [8]. In proliferating cells such as mouse embryonic fibroblasts (MEFs) and cancer cells, mTORC1 controls transcription of glycolysis, lipid and lysosome biogenesis, and mitochondrial metabolism genes [1,3] and also activates several transcription factors involved in metabolic processes including sterol regulatory element-binding proteins (SREBPs) and hypoxia-inducible factor-1α (HIF1α) [1,9,10,11].

The dysregulation of transcriptional processes most likely contributes to disease development in neurodevelopmental disorders associated with changes in mTOR activity —yet mTOR-mediated transcriptional regulation in the brain has been poorly investigated.

In the brain, the mTOR signaling pathway controls various cellular processes such as neuronal differentiation, neuronal cell size determination, axon guidance, dendritogenesis, and synaptic plasticity [12]. Dysregulation of the mTOR pathway has been associated with epilepsy, ASD, and intellectual disability (ID). In this context, both hyper- and hypoactive mTOR signaling are connected to disease. mTOR hyperactivity was observed for example in fragile X syndrome (FXS), neurofibromatosis 1 (NF1), tuberous sclerosis, and Opitz BBB/G syndrome. mTOR hypoactivity was one characteristic of a mouse model for Rett syndrome. It was also observed in neurons that were derived from human embryonic stem cells (hESC) and carried a *MECP2* loss-of-function allele [13,14,15,16,17,18].

mTOR activity is regulated intracellularly by nutrients, energy level, and stress factors (e.g., hypoxia) and extracellularly by growth factors (e.g., brain-derived neurotrophic factor (BDNF)), hormones (e.g., insulin), neurotransmitters, and cytokines. Upon activation, mTORC1 phosphorylates eukaryotic translation initiation factor 4E (eIF4E) binding proteins (4E-BP1 and 4E-BP2) and S6 kinases (S6K1 and S6K2) which influence different steps of translation [8,19]. Two studies showed that, rather than having a global effect on protein translation, mTOR inhibition specifically affects translation of mRNAs containing a 5′-terminal oligopyrimidine tract (5′TOP) or a pyrimidine-rich translational element (PRTE), in their 5′-untranslated regions (5′UTRs) [20,21]. While protein translation regulation by mTOR has been studied extensively in the brain, much less is known about mTOR-mediated transcriptional regulation.

Here, we used 3′mRNA sequencing (3′mRNA-Seq) to identify mTOR-dependent genes, the expression of which is changed upon mTOR inhibition in murine neurons. We found that using the mTOR inhibitor temsirolimus [22] numerous genes of the sterol/cholesterol biosynthesis pathway were downregulated, which resulted in decreased cholesterol levels.

Injection of rapamycin into pregnant dams confirmed our results in vivo, in the developing pre- and postnatal cerebral cortex. In summary, our findings strongly suggest an important role for the mTOR pathway in regulating the expression of genes involved in lipid metabolism in neurons and the developing brain.

## 2. Results

### 2.1. Transcriptional Targets Downstream of mTOR in Neurons

To identify mTOR-dependent transcriptional changes, we treated primary cortical neurons with the mTOR inhibitor temsirolimus. Temsirolimus treatment had a strong effect on mTOR activity as measured by the ratio of the mTOR downstream effector pS6/S6 (Figure 1A). With the mock-treated, and the 5 h and 24 h treated cells, we carried out 3′mRNA-Seq and identified a total of 522 differentially expressed genes (DEGs) at 5 h and 1090 DEGs at 24 h of temsirolimus treatment with an overlap of 241 DEGs between the two time points (0.5 log2 fold change and adjusted *p*-value < 0.05). Of all DEGs 315 and 564 were upregulated at 5 and 24 h, respectively, and 207 and 526 were downregulated (Figure 1B,C, Appendix A).

Because mTOR is best known for its function in gene activation, we decided to focus on the set of genes downregulated upon mTOR inhibition in subsequent analyses. mRNA expression changes can be mediated at the epigenetic level, by changes in chromatin accessibility, at the transcriptional level, by changes in transcription factor binding, or at the posttranscriptional level, by changes in mRNA stability. It was demonstrated before that mTOR activation can both enhance transcription of metabolic genes and increase their mRNA stability [1,6]. To test for possible mRNA stability changes we treated primary neurons with the transcriptional inhibitor Actinomycin D in combination with temsirolimus or DMSO and measured the mRNA decay rate of three downregulated targets, Ldlr, Osc, and Nsdhl. None of the three mRNAs underwent a decrease in stability upon mTOR inhibition, which suggests that posttranscriptional regulation of the mRNA amount is not a general mechanism of gene regulation under these experimental conditions (Appendix A). 

To test for mTOR-mediated chromatin accessibility changes, we employed an assay for Transposase-Accessible Chromatin with high-throughput sequencing (ATAC-seq) on biological replicates (*n* = 2 for each condition) from cortical neurons treated with temsirolimus or DMSO for 5 h. We identified 62,225 and 53,051 (Replicate 1) and 119,339 and 90,435 (Replicate 2) open chromatin regions in DMSO- and temsirolimus-treated neurons, respectively (*p* < 0.05; Appendix A). The majority of open chromatin regions mapped to promoters, introns, and intergenic regions of annotated genes (Appendix A). When comparing open chromatin regions of DMSO- and temsirolimus-treated neurons, we could, however, not identify major differences, especially not in those regions associated with genes involved in cholesterol biosynthesis (Appendix A). This suggests that modifications in chromatin accessibility do not play a major role in mediating gene expression changes after temsirolimus treatment. Rather, this suggests that mTOR-mediated activation of gene expression in neurons occurs directly at the level of transcription factor regulation. Several studies have reported mTOR-regulated transcriptional changes in non-neuronal cells. For instance, Duvel et al. [1] used *Tsc1*−/− and *Tsc2*−/− MEFs, which exhibit growth factor independent activation of mTORC1 in combination with rapamycin treatment. The authors found that mTORC1 activates the expression of numerous genes involved in glycolysis, the pentose phosphate pathway, and lipid/sterol biosynthesis. We used this dataset to identify mTOR-dependent transcriptional changes shared between MEFs and neurons. Surprisingly, when we compared our set of downregulated genes with the 130 genes found to be upregulated by mTORC1 in MEFs, only 13 genes overlapped between both sets after 5 h of temsirolimus treatment and only 20 genes after 24 h (Figure 1D, Appendix A).

### 2.2. mTOR Regulates the Expression of Genes of the Cholesterol Pathway in Primary Neurons

Among the genes overlapping in neurons and MEFs were genes of the cholesterol synthesis pathway. Gene ontology analysis of our neuronal data set revealed that genes downregulated after 5 or 24 h of temsirolimus treatment were enriched for metabolic terms (Figure 2A). These terms comprised sterol/lipid biosynthesis/metabolism processes including the cholesterol biosynthesis pathway (Figure 2A). Other metabolic pathways previously described to be dependent on mTOR signaling such as the glycolysis and pentose phosphate pathway [1] were, however, not enriched. By qRT-PCR analysis of gene expression changes, we could confirm a general downregulation of sterol/cholesterol pathway genes. Genes of the glycolysis pathway, however, remained unaltered in their expression (Figure 2B). A 40–60% downregulation of genes involved in the cholesterol biosynthesis process was found after 5 h of temsirolimus treatment (Figure 2C). Western blot experiments confirmed significant downregulation of Ldlr and Osc proteins after 24 h of temsirolimus treatment and for Mvd a trend of downregulation was seen (*p* = 0.13; Figure 2D, pS6/S6 ratio served as proof of efficacy of treatment). In contrast to genes of the cholesterol biosynthesis pathway, genes of the glycolysis pathway (*Pfkp*, *Pgm2*, and *Pdk1*) remained unaltered (Figure 2C). Temsirolimus is a derivative of rapamycin, an interaction partner of FK506-binding protein-12 (FKBP-12) and allosteric partial inhibitor of the mTORC1 kinase [22]. In contrast to rapamycin (and temsirolimus), ATP-competitive inhibitors such as Torin 1 inhibit the phosphorylation of all mTORC1 substrates. The observed effect that temsirolimus treatment caused inhibition of the genes of some, but not all metabolic pathways previously described to be dependent on mTOR activity, could theoretically be due to temsirolimus being only a partial mTORC1 inhibitor. Therefore, we treated neurons with Torin 1 for 5 h. As expected, Torin 1 treatment not only caused a strong reduction of pS6, but also of p4EBP (Appendix A). Like temsirolimus treatment, the cholesterol biosynthesis pathway genes *Ldlr* and *Dhcr7* were significantly downregulated after 5 h of Torin 1 treatment (Appendix A). In contrast to temsirolimus treatment; however, the glycolysis pathway genes *Pfkp*, *Pgm2*, and *Pdk1* were also significantly downregulated (Appendix A). 

### 2.3. mTOR Activity Is Essential for Proper Expression of Cholesterol Pathway Genes in the Embryonic and Postnatal Cerebral Cortex

Having shown that mTOR promotes the expression of cholesterol pathway genes in primary neurons in vitro we next wanted to validate this in the brain in vivo. While neurons of the adult brain rely mainly on astrocytes for cholesterol, in a critical time window during development, neuronal cholesterol synthesis is essential for neurons to differentiate normally [23]. To analyze if mTOR activity is required for the expression of cholesterol pathway genes we, therefore, chose to inhibit mTOR prenatally starting at E16.5, by injecting rapamycin which has the same mechanism of action as temsirolimus intraperitoneally into pregnant mice. Twenty-four hours after a single rapamycin injection we observed a strong inhibition of the mTOR pathway as shown by Western blot analysis of the mTOR downstream effector pS6 (Figure 3A). Already at this time point, RT-qPCR showed that all four tested cholesterol pathway genes (*Ldlr*, *Osc*, *Mvd*, *Dhcr7*) were reduced in their mRNA expression by about 50% (Figure 3B). In contrast, only *Ldlr* was significantly reduced by about 20% at the protein level at this time point (Figure 3C). Because the low effect on protein level could be due to high protein stability, we subsequently injected rapamycin on three consecutive days. Surprisingly, after three days of injection, mRNA levels of the four tested genes had returned to their normal levels which might be due to compensatory effects (Figure 3D). Protein expression of Ldlr, Mvd, and Nsdhl was, however, significantly reduced by 45, 30, and 25%, respectively (Figure 3E). 

To further corroborate our data, we repeated the i. p. injections of DMSO and rapamycin at early postnatal stages starting at postnatal day two (P2). A single injection of rapamycin at this stage led to similar results as a single injection at prenatal stage E16.5 (Figure 4A–C). Twenty-four hours after rapamycin injection, the mTOR pathway was strongly inhibited and all three genes analyzed, *Ldlr*, *Osc*, and *Mvd* were significantly downregulated at the mRNA level (Figure 4B). As seen after a single prenatal rapamycin injection, only *Ldlr* was also downregulated at the protein level under these conditions (Figure 4C). To test for a delayed reduction in protein levels, we injected rapamycin twice (on P2 and P4) and analyzed the mice at P6. Such a prolonged mTOR inhibition caused overall growth retardation in the rapamycin-injected mice when compared to DMSO-injected mice. Thus, rapamycin-injected mice were smaller and weighed less than DMSO-injected mice—an effect that was highly significant (Figure 4D). Similar to three days of prenatal rapamycin injection, mRNA levels of Ldlr, Osc, and Mvd had returned to normal levels, and instead, protein levels of Ldlr, Mvd, and Nsdhl were significantly reduced by about 40–50% (Figure 4E,F).

### 2.4. mTOR Inhibition Reduces Cholesterol Production In Vitro and In Vivo

After 5 and 24 h of temsirolimus treatment, eight and fifteen genes of the cholesterol biosynthesis pathway, respectively, were significantly downregulated in their expression (Figure 5A, Appendix A). The cholesterol biosynthesis pathway can be separated into three sections according to the type of compounds that are produced—mevalonate, isoprenoids, and sterols. After 24 h of temsirolimus treatment, genes of all three sections were downregulated in their expression, including the gene encoding Hmgcr which catalyzes the rate-limiting step in cholesterol biosynthesis; the reduction of HMG-CoA to mevalonate.

We, therefore, expected impaired cholesterol biosynthesis in neurons treated with temsirolimus. To test for this, we treated cortical neurons with temsirolimus for two days and performed Amplex Red cholesterol assays. Compared with DMSO treated neurons, cholesterol levels were significantly reduced in temsirolimus-treated neurons as well as in neurons treated with cyclodextrin, a cholesterol-depleting reagent which served as a positive control (Figure 5B).

To further corroborate our data and to test for their in vivo relevance we measured cholesterol levels in the brains of rapamycin-injected mouse embryos. In agreement with the results obtained in neurons, we saw a significant decrease in cholesterol production in the embryonic cerebral cortex after injection of the mTOR inhibitor (Figure 5C).

### 2.5. mTOR Dependent Genes Contain SREBP, NF-YA, and SP1 Binding Sites in Their Promoter Regions

Düvel and colleagues [1] showed that in MEFs the binding site for SREBPs is over-represented in the promoters of mTOR-regulated targets. It is also known that mTORC1 regulates the nuclear abundance of SREBP1 and that SREBP transcription factors are required for mTOR-induced expression of metabolic genes [1,9]. When using the software findM [24] to screen the promoter regions of the genes downregulated in neurons treated with temsirolimus for 5 h we found that 44.9% of them contained at least one such binding site (Figure 6A).

We applied the software Homer [25] on 5 h temsirolimus vs. DMSO downregulated genes to identify additional transcription factor binding sites in an unbiased manner. This analysis revealed that the binding sites for the transcription factors SP1 and NF-Y, respectively, occurred in 56 and 52% of the promoters of the downregulated genes (Figure 6B,C). SREBPs, which are weak transcriptional activators on their own [26], cooperate with other transcription factors on their target promoters. Two key interaction partners of SREBP1 are SP1 and NF-Y [27]. In our data, we found that of the 93 downregulated genes that contained an SREBP binding motif in their promoters, 83 (89.2%) also contained an SP1 motif and 38 (40.9%) an NF-Y motif (Figure 6D). Interestingly, the majority (36 out of 38) of downregulated genes containing both an SREBP and an NF-Y binding site also had an SP1 motif in their promoters (Figure 6D).

Identification of enriched GO terms revealed that genes involved in the sterol/cholesterol biosynthesis pathway preferentially contained binding sites for all three (SREBP, NF-Y, SP1) or two (NF-Y, SP1) of the three factors in their promoters (Figure 6E).

## 3. Discussion

Surprisingly, it has not yet been investigated whether mTOR controls the transcription of metabolic genes in neuronal cells in the brain.

Previously, it was shown that the mTOR kinase is an important transcriptional activator of metabolic genes in non-neuronal cellular systems such as MEFs, regulatory T cells, and cancer cell lines [1,3,5]. Having in mind that mTOR signaling is an essential pathway during brain development and that mTOR dysregulation causes various neurodevelopmental disorders, dysregulation of metabolic gene expression may well contribute to neurological symptoms in mTOR-associated disorders. Here we show that mTOR drives the expression of metabolic genes of the sterol/cholesterol pathway in primary cortical neurons in vitro and in the developing cerebral cortex in vivo.

Although in neurons like in non-neuronal cells the mTOR kinase promotes the expression of metabolic genes, we observed a major difference between the different cell types: in neurons treated with temsirolimus, we found downregulation of the cholesterol biosynthesis pathway genes but not of the glycolysis and pentose phosphate pathways which were both affected in MEFs [1]. One explanation for this discrepancy might be the different experimental conditions. Indeed, when we treated cortical neurons with the mTOR inhibitor, Torin 1, which inhibits the phosphorylation of mTORC1 substrates more completely than rapamycin and temsirolimus, we not only observed downregulation of genes of the cholesterol biosynthesis pathway, but also of the glycolysis pathway. Because Torin 1 does not cross the blood-brain barrier we could not test for this effect in vivo in the same way as we did in vitro. Besides, prior studies suggest that cell type-specific effects exist which are likely to contribute to the differences observed between non-neuronal cell types and neurons. For example, depending on the cell type, mTOR activates specific transcription factors such as HIF1α, SREBPs, and TFEB in MEFs and IRF4 and GATA3 in regulatory T cells [1,3,28]. Even nuclear localization and binding of mTOR itself to DNA were reported [5,10,29]. Which mechanisms contribute to the regulation of cholesterol gene expression by mTOR in neurons is still unclear. We found a significant enrichment of binding sites for the transcription factors SP1, SREBPs, and NF-Y in the promoters of mTOR-regulated genes in neurons. To our knowledge, a binding sites’ enrichment for NF-Y has not yet been described in promoters that respond to mTOR activity. NF-Y regulates an increasingly identified number of genes and is ubiquitously expressed, so it is likely to have different roles depending on the cellular context. The NF-YA knockout in mice for example, causes early embryonic lethality [30], underpinning its importance for physiological functions. In proliferating cells such as MEFs and hematopoietic stem cells, NF-Y is involved in cell cycle regulation [31,32,33,34]. While it is downregulated in most tissues during differentiation [35,36], NF-Y is active in mature neurons of the adult mouse brain where its deletion causes neurodegeneration [37,38]. NF-Y co-localizes with other transcription factors such as Fos proto-oncogene (FOS) at genomic sites [39]. In differentiating neurons and HEK293 cells, NF-YA and c-Jun N-terminal kinase (JNK) were found to bind to the same genomic sites and NF-YA recruits JNK to these sites [40]. It was also shown that NF-Y interacts at target gene promoters in cooperation with SREBP1 or SREBP2 and SP1 and activates genes of metabolic pathways [26,41,42].

Downregulation of mTOR activity is thought to play an essential role in the pathogenesis of neurodevelopmental disorders such as Rett syndrome and Cyclin-Dependent Kinase Like 5 (CDKL5) deficiency disorder [14,43,44]. How reduced mTOR activity contributes to disease development in these disorders is, however, not entirely clear. Defects in the cholesterol pathway may be a contributing mechanism. This is supported by the observation that even small perturbations in cholesterol metabolism can primarily affect neuronal development [45,46,47,48]. Low levels of cholesterol have been associated with a variety of neurodevelopmental disorders. Cholesterol biosynthesis is a multistep process divided into a pre- and post-squalene pathway. Low levels of cholesterol have been associated with a variety of neurodevelopmental disorders and several genes of the cholesterol biosynthesis pathway have been found to carry mutations. The most common genetic syndrome associated with defects in cholesterol biosynthesis is the autosomal recessive Smith Lemli Opitz syndrome (SLOS; OMIM# 270400) which is caused by mutations in *DHCR7* encoding the enzyme 7-dehydrocholesterol D7reductase [49,50]. The neurological symptoms of SLOS include epilepsy, ID, and behavioral problems, among others. Mutations in NAD(P) steroid dependent dehydrogenase-like (*NSDHL*) gene are associated with the X-linked dominant disorder CHILD syndrome (OMIM #308050; [51]) and the X-linked recessive disorder CK syndrome (OMIM #300831; [52]), and mutations in mevalonate kinase (*MVK*) with the autosomal recessive disorder Mevalonic Aciduria (OMIM #610377; [53]). All three syndromes are characterized by structural brain abnormalities and/or neurological symptoms including ID. The gene product of *NSDHL*, 3β-hydroxysteroid dehydrogenase, is involved in one of the later steps in cholesterol biosynthesis, and the gene product of *MVK*, mevalonate kinase, is a peroxisomal enzyme involved in cholesterol biosynthesis in the pre-squalene pathway. Expression of all three genes, *Dhcr7*, *Nsdhl*, and *Mvk* were found downregulated in temsirolimus-treated neurons. Although these results suggest that perturbations in cholesterol biosynthesis in disorders associated with mTOR downregulation may contribute to disease development, it is still unclear whether mTOR downregulation leads to a long-lasting reduction of cholesterol levels in the brain. In the in vivo experiments, we observed that 24 h after injecting a single dose of rapamycin the mRNA expression of all tested genes was decreased. After three days (prenatal) or four days (postnatal) of mTOR inhibition, however, it had returned to normal levels. In contrast, protein levels of most tested genes only decreased significantly after three (prenatal) or four days (postnatal) of mTOR inhibition. Even more surprising, OSC protein expression did not change when a single dose of rapamycin was injected prenatally but increased significantly when a single dose of rapamycin was injected postnatally. The delayed reduction at the protein level could possibly be caused by enhanced protein stability. In addition, these observations hint at feedback and compensatory mechanisms, which is confirmed in a study by Buchovecky and colleagues [54] who found that in the brains of a mouse model for Rett syndrome (*Mecp2* null mice) total cholesterol was increased at P56 when mutant males had developed severe symptoms. At a later age (P70), however, brain cholesterol levels were comparable to wildtype levels which reflected reduced cholesterol synthesis. Likely, the overproduction of cholesterol had fed back to a decreased cholesterol synthesis later. Genetic or pharmacologic inhibition of cholesterol synthesis ameliorated some of the symptoms in these mice. This suggests a critical window during brain development when neuronal tissue depends particularly on neuronal cholesterol biosynthesis.

Our studies were focused on mTOR-mediated regulation of cholesterol biosynthesis in cortical neurons. Because the brain needs to synthesize its own cholesterol even during development when the blood-brain barrier has not fully formed yet, neuronal cholesterol synthesis is most important during a critical developmental time window [23]. Neurons in the adult brain rely mainly on astrocytes for providing cholesterol. Also, during myelination oligodendrocytes synthesize vast amounts of cholesterol. It was already shown in zebrafish that cholesterol is needed for mTOR activity in oligodendrocyte precursor cells and that mTOR regulates cholesterol-dependent myelin gene expression [55]. Therefore, an open question is whether mTOR signaling in astrocytes and oligodendrocytes is equally essential for cholesterol biosynthesis as it is in neurons.

## 4. Materials and Methods

### 4.1. Mice, Cell Culture, and Drug Treatment

NMRI mice (8–12 weeks old) were obtained from Janvier labs (Saint Berthevin, France) and sacrificed by cervical dislocation. For neuron culture, primary cortical neurons were isolated from E14.5 embryos from pregnant NMRI mice. After the collection of brains, cortices were dissected and mechanically separated into single cells via resuspension. The neurons were plated on Poly-L-Ornithine (Sigma, St. Louis, MO, USA) and Laminin (Sigma St. Louis, MO, USA)-coated plates, and cultured in Neurobasal medium (Gibco, Carlsbad, CA, USA), supplemented with 2% B-27 plus vitamin A (Gibco, Carlsbad, CA, USA) and 1% GlutaMAX (Gibco, Carlsbad, CA, USA), in a humidified incubator at 37 °C and 8% CO_2_. For drug treatment, primary cortical neurons were cultured for six days in 6-well-plates followed by treatment with the mTOR inhibitor temsirolimus (10 µM in DMSO; Sigma, St. Louis, MO, USA) or Torin 1 (500 nM in DMSO; LC Laboratories, Woburn, MA, USA) diluted in culture medium for the indicated periods. For measuring RNA stability, neurons were treated with the transcriptional blocker Actinomycin D (Sigma, St. Louis, MO, USA; 5 µg/mL). For further analyses, cells were harvested in PBS using cell scrapers. For in vivo experiments in prenatal mice 8–12 weeks old, pregnant NMRI mice received a single intraperitoneal (i. p.) injection at stage E16.5 or were injected i. p. once daily with rapamycin (LC Laboratories, Woburn, MA, USA) or DMSO (control vehicle) on three consecutive days. For in vivo experiments in postnatal mice, two day old mice received a single i. p. injection or were injected i. p. twice at P2 and P4. A single injection contained either 10 µL DMSO or 10 µL rapamycin (1 mg/kg) dissolved in 100% DMSO. Twenty-four to 48 h after the last injection, embryonic cortices were isolated and stored either in lysis buffer (48% Urea, 14 mM Tris pH 7.5, 8.7% Glycerol, 1% SDS, protease inhibitor cocktail (Roche, Basel, CH, Switzerland)) or RNAlater (Sigma, St. Louis, MO, USA) for further analysis at −80 °C.

### 4.2. Immunoblotting

Antibodies for immunoblotting were as follows: phospho-S6, S6 (Cell Signaling Technology, Danvers, MA, USA), Osc, Mvd (Santa Cruz Biotechnology, Santa Cruz, CA, USA), Ldlr, Nsdhl (Novus Biologicals, Centennial, WY, USA), GAPDH (Abcam, Cambridge, UK). Western blot analysis was performed by standard methods using enhanced chemiluminescence.

### 4.3. RNA Isolation, cDNA Synthesis, RT-qPCR, and RNA Sequencing

Total RNA extraction with TRIzol reagent from brain tissue was performed as recommended by Invitrogen Life Technologies (Carlsbad, CA, USA). A High Pure RNA Isolation Kit (Roche, Basel, CH, Switzerland) was used to extract total RNA from cell samples using spin columns. The purity, quantity, and integrity of the RNA were measured with a NanoDrop One spectrophotometer (Thermo Fisher Scientific, Waltham, MA, USA). According to the manufacturer’s instructions, the cDNA samples were synthesized from 500 to 1000 ng total RNA using the PrimeScriptTM RT Master Mix cDNA (Takara, Kyoto, Japan). Quantitative real-time PCR (qRT-PCR) was carried out using SYBR^®^ Premix Ex Taq™ II (Tli RNaseH Plus) and 10 μM primers (final concentration), according to the manufacturer’s instructions. RT-qPCR reactions were performed on an ABI StepOnePlus Real-Time PCR System using intron spanning primers with the following conditions: 95 °C/30 s, 40 cycles of 95 °C/5 s, 60 °C/30 s, 72 °C/30 s. For primer sequences see Appendix A. All reactions were measured in triplicate, and median cycles to threshold (Ct) values were used for analysis. The housekeeping gene *GAPDH* was used for normalization, and relative gene expression was determined using the 2−ΔΔCT method.

For RNA sequencing, RNA purity and integrity were measured using the Agilent 2100 bioanalyzer (Agilent Technologies, Santa Clara, CA, USA). RNA was converted into cDNA by using a QuantSeq 3′mRNA-Seq Reverse (REV) Library Prep Kit (Lexogen, Vienna, Austria) according to the manufacturer’s instruction to generate a compatible library (2.5 pM) for Illumina sequencing. RNA sequencing was performed using a high output reagent cartridge v2 (75 cycles; Illumina, San Diego, CA, USA) with a custom primer (0.3 µM) provided by Lexogen on an Illumina NextSeq 500 device.

### 4.4. RNA-Seq Data Analysis

After the sequencing, bcl2fastq v2.17.1.14 conversion software (Illumina, San Diego, CA, USA) was used to demultiplex sequence data and convert base call (BCL) files into Fastq files. Sequencing adapters (AGATCGGAAGAG) were trimmed and reads shorter than six nucleotides were removed from further analysis using Cutadapt v1.11 [56]. Quality control checks were performed on the trimmed data with FastQC v0.11.4 [57]. Read mapping of the trimmed data against the mouse reference genome and transcriptome (mm9) was conducted using STAR aligner v2.5.3 [58]. To estimate each transcript’s expression levels the mapped reads were assigned to annotated features using the Subread tool featureCounts v1.5.2 [59]. The output of the raw read counts from featureCounts was used as an input for the differential expression analysis using the combination of DESeq2 v1.16.1 [60] and edgeR v3.26.8 [61] packages. Pairwise comparison analysis of two different conditions was carried out with edgeR to normalize the expression levels of known genes. Only genes with CPM (counts per million) > 10 were further analyzed. The differentially expressed gene analysis was performed on the normalized genes’ expression using DESeq2 with |log2FoldChange| > 0.5 and padj (adjusted *p*-value) > 0.05. Overlapping DEGs were visualized using VennDiagram package v1.6 [62] in R. Heatmaps of up- and downregulated genes was created using ComplexHeatmap package v2.7 [63] in R.

### 4.5. ATAC-Seq

ATAC-seq was done as previously described [64]. Briefly, 50,000 DMSO- or temsirolimus-treated cells were resuspended in cold lysis buffer (10 mM Tris-HCl, pH 7.4, 10 mM NaCl, 3 mM MgCl_2_ and 0.1% IGEPAL CA-630) and centrifuged at 750 g for 30 min at 4 °C. Immediately following the cell preparation, the pellet was resuspended in the transposase reaction mix (25 μL 2× TD buffer, 2.5 μL transposase (Illumina, San Diego, CA, USA) and 22.5 μL nuclease-free water). The transposition reaction was carried out for 30 min at 37 °C. Following transposition, the sample was purified using a Qiagen MinElute kit (Qiagen, Hilden, Germany). Following purification, the library was amplified using 1× NEBnext PCR master mix and 1.25 μM of custom Nextera PCR primers 1 and 2 [64], using the following PCR conditions: 72 °C for 5 min; 98 °C for 30 s; and thermocycling at 98 °C for 10 s, 63 °C for 30 s and 72 °C for 1 min. After 11–12 cycles of PCR amplification, the sample was further purified using Qiagen MinElute kit (Qiagen, Hilden, Germany). To remove primer dimers from the samples they were further purified using AMPure beads XP (Beckman Coulter, Brea, CA, USA) with a ratio of x0.9 of beads to samples. Samples were then analyzed in a bioanalyzer (Agilent Technologies, Santa Clara, CA, USA) and sequenced on an Illumina NextSeq 500 Illumina, San Diego, CA, USA.

### 4.6. ATAC-Seq Data Analysis

ATAC-Seq data quality check was performed using reads FASTQC v0.11.8 [56]. Further, adaptors were removed using Trimmomatic v0.39 [65]. Paired-end ATAC-Seq reads were mapped to Mus musculus genome (mm10) UCSC annotations using Bowtie2 v2.3.5.1 [66] with default parameters. Properly paired-end reads with high mapping quality (MAPQ ≥ 10) were retained in analysis with Samtools v1.7 [67]. Next, using Picard tools MarkDuplicates [68] utility duplicates were removed. ATAC-Seq peaks were called using MACS2 v2.1.1.20160309 [69] and were visualized with the USCS genome browser [70].

### 4.7. Cholesterol Assay

Primary cortical neurons were treated with DMSO or temsirolimus at 4 days in vitro (DIV4) for 48 h and with Cyclodextrin (Sigma, St. Louis, MO, USA) for one hour before cholesterol measurement. The cells were harvested using cell scrapers in a volume of 500 µL DPBS. The AmplexTM Red Cholesterol Assay (Thermo Fisher Scientific, Waltham, MA, USA) was performed in a black 96 well plate by the reaction of 50 µL of Amplex Red working solution with 50 µL of assay sample. A 5 mL volume of working solution, prepared before the analysis, contained 75 µL of the Amplex Red reagent (300 µM) and 2 U/mL of HRP (horseradish peroxidase), 2 U/mL cholesterol oxidase, and 0.2 U/mL cholesterol esterase. The working solution volume was adjusted to 5 mL with reaction buffer, which contained 25 mM potassium phosphate, pH 7.4, 12.5 mM NaCl, 1.25 mM cholic acid, and 0.025% TritonX-100. The reactions were incubated for 30 min at 37 °C, protected from light. After incubation, fluorescence was measured in a fluorescence microplate reader FLUOstar Optima (BMG Labtech, Ortenberg, Germany) using an excitation wavelength of 560 nm and emission detection at 590 nm.

For in vivo cholesterol measurement, cortex samples were isolated after three injections on three consecutive days (E14.5–E16.5) of DMSO or rapamycin (1 mg/kg) and collected in cholesterol lysis buffer (50 mM Tris-HCl pH 7.5, 150 mM NaCl, 1% Nonidet-P40, 0.5% deoxycholic acid, Protease Inhibitor Cocktail). The AmplexTM Red Cholesterol Assay (Thermo Fisher Scientific, Waltham, MA, USA) was performed in Qubit tubes using the Qubit 2.0 Fluorometer. Cortex samples were diluted by using 20 µL cortex samples and 80 µL 1× reaction buffer and transferred into Qubit tubes. After adding 100 µL Amplex Red working solution (see above) to each tube the samples were incubated for 15 min at room temperature, protected from light. Subsequently, the reaction was stopped using AmplexTM Red/UltraRed Stop Reagent (Thermo Fisher Scientific, Waltham, MA, USA) and fluorescence was measured using a Qubit 2.0 Fluorometer.

### 4.8. Statistical Analyses

All statistical analyses except for NGS analyses were done with GraphPad Prism 5 or Microsoft Office Excel version 16.49. Data are shown as mean + standard error of the mean. Statistical analyses were done using Student’s *t*-test. *p* values < 0.05 were considered statistically significant.

### 4.9. Gene Ontology Analysis and KEGG Pathway Analysis

For differentially up- and down-regulated expressed genes of each pairwise comparison as well as overlapping genes for SREBP, NFY, and SP1 motifs, an overrepresentation analysis (ORA) was carried out with clusterProfiler v3.4.4 [71]. All expressed genes within the pairwise comparison samples with CPM > 10 served as a background for the analysis. The Bioconductor org.Mm.eg.db v 3.8.2 mouse annotation package [72] and mouse KEGG.db [73] were used for the gene ontology analysis and KEGG analysis, respectively. The default parameters were used for all over-representation tests.

### 4.10. Motif Identification

Motif discovery analysis within promoter sequences was performed using HOMER (Hypergeometric Optimization of Motif EnRichment) Software v4.9 [25]. ATAC-Seq sequencing data from 186 genes in FASTA format served as input for the findMotif.pl function. Scrambled input sequences (randomized) were created automatically by HOMER and used as a background for the motif analysis.

## Figures and Tables

**Figure 1 ijms-22-06034-f001:**
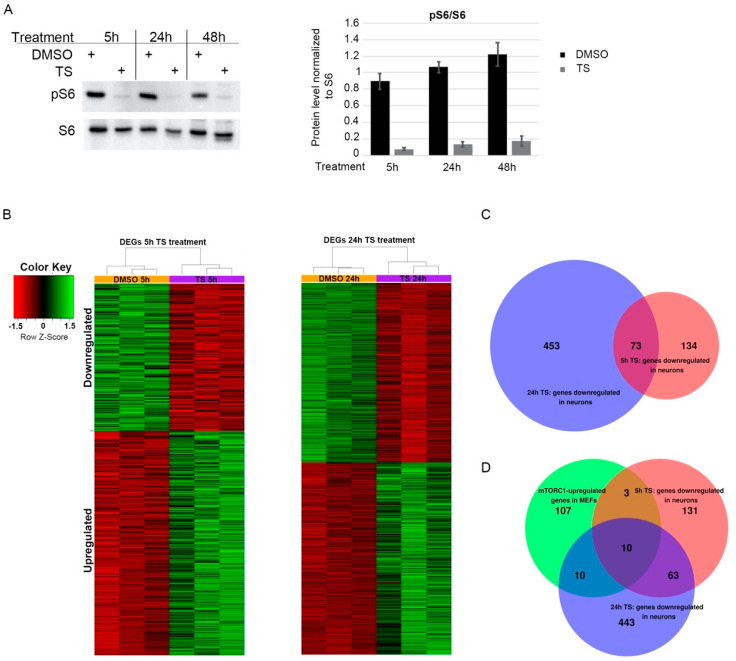
Temsirolimus treatment leads to widespread gene expression changes in primary neurons. Primary cortical neurons at days in vitro (DIV) 6 were treated with temsirolimus (10 µM) or vehicle (DMSO) and subjected to Western blot analysis (**A**) or RNA-Seq (**B**,**C**). (**A**) The cells were lysed at the indicated time points and subjected to immunoblot analysis with pS6 and S6 specific antibodies. Bands were quantified using ChemiDoc software (Image Lab V5. Version 5.2.1.). (**B**–**D**) For RNA-Seq analysis, total RNA was extracted from DMSO and temsirolimus-treated neurons from three different biological replicates each and converted into cDNA using a QuantSeq 3′mRNA-Seq Reverse (REV) Library Prep Kit (Lexogen, Vienna, Austria). (**B**) Heat maps indicating differentially expressed genes (DEGs) between neurons treated with DMSO or temsirolimus for 5 or 24 h. (**C**) Venn diagram of genes downregulated after 5 or 24 h of temsirolimus treatment. (**D**) Venn diagram showing the overlap between genes downregulated in neurons after 5 and 24 h of temsirolimus treatment and upregulated in mTOR hyperactive MEFs [1].

**Figure 2 ijms-22-06034-f002:**
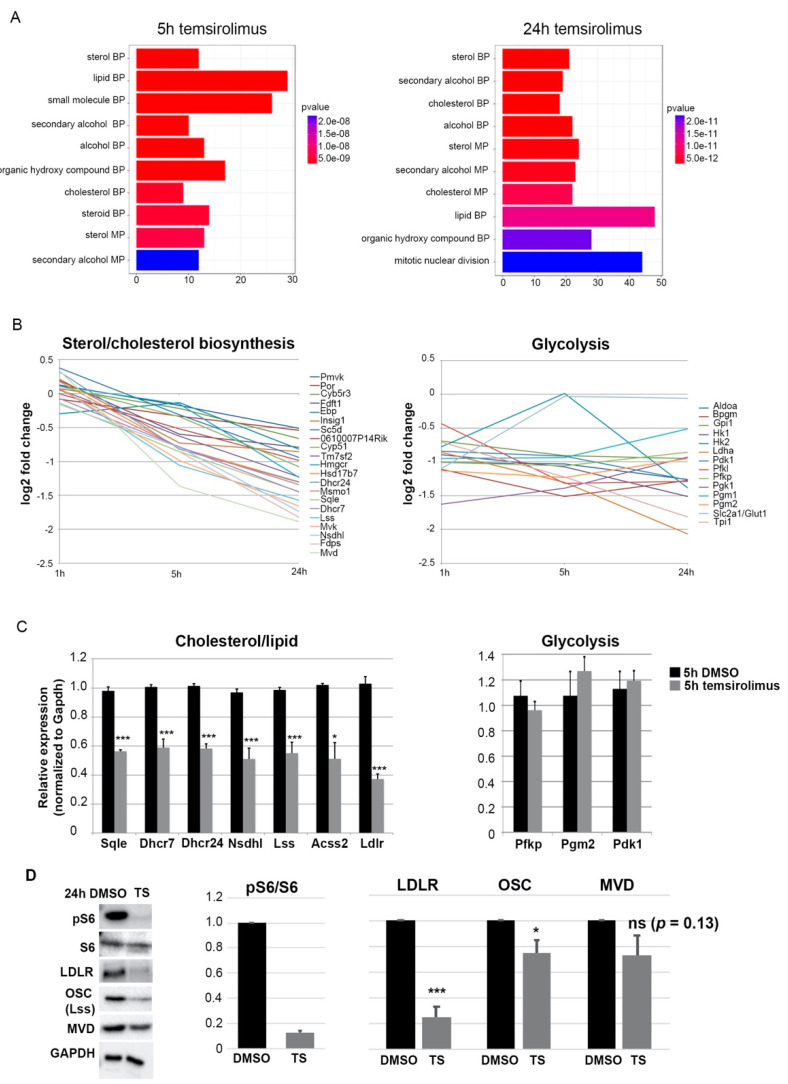
Temsirolimus treatment causes downregulation of genes of the sterol/cholesterol biosynthesis pathway. (**A**) Barplots of the top 10 GO enriched terms of downregulated genes after 5 h (left) and 24 h (right) of temsirolimus treatment. Both plots show a high enrichment for metabolic pathways shown on the *y*-axis and the number of gene counts on the *x*-axis. Enrichment scores are depicted in the colored bar. (**B**) Inhibition of mTORC1 by temsirolimus treatment downregulates the sterol and cholesterol biosynthesis pathway genes but does not change expression of glycolysis genes. (**C**) Confirmation of changes in the expression of cholesterol/glycolysis genes by RT-qPCR. Cells were lysed after 5 h of temsirolimus/DMSO treatment and subjected to total RNA extraction and RT-qPCR; mRNA expression was normalized to the housekeeping gene *GAPDH* (**D**) Western blot experiments in primary neurons after treatment with DMSO or temsirolimus for 24 h. Reduction in pS6 in relation to S6 confirms mTORC1 inhibition. Expression pattern of proteins involved in cholesterol biosynthesis. Data represent the average of three biological replicates (from three independent donor litters). For statistical analyses, Student’s *t*-test was used. * *p* < 0.05; *** *p* < 0.001.

**Figure 3 ijms-22-06034-f003:**
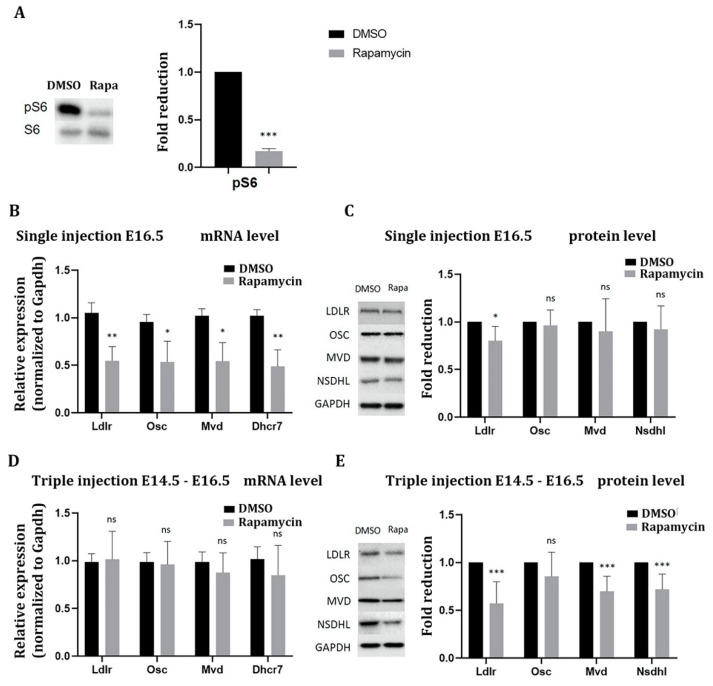
mTORC1 inhibition in the embryonic cerebral cortex downregulates cholesterol biosynthesis genes in vivo. Pregnant mice received either a single injection according to E16.5 of the offspring or were injected on three consecutive days (E14.5-E16.5) once per day with DMSO or rapamycin at a dose of 1 mg/kg. Subsequently, total RNA and protein were extracted from the cerebral cortex and RT-qPCR and Western blot experiments were performed. (**A**) Inhibition of mTORC1 activity was confirmed by Western blot analysis of the mTORC1 downstream effector pS6 (compared to S6) 24 h after a single rapamycin injection. (**B**,**D**) RT-qPCR to detect RNA expression of the four genes *Ldlr*, *Osc*, *Mvd*, and *Dhcr7* after one day (**B**) or three days (**D**) of i. p. injection of rapamycin compared to DMSO; mRNA expression was normalized to Gapdh. (**C**,**E**) Immunoblot analysis to detect Ldlr, Osc, Mvd, and Dhcr7 protein expression after one day (**C**) or three days (**E**) of i. p. injection of rapamycin compared to DMSO and normalized to Gapdh. Data represent the average of six biological replicates. For statistical analyses, Student’s *t*-test was used. * *p* < 0.05; ** *p* < 0.01; *** *p* < 0.001.

**Figure 4 ijms-22-06034-f004:**
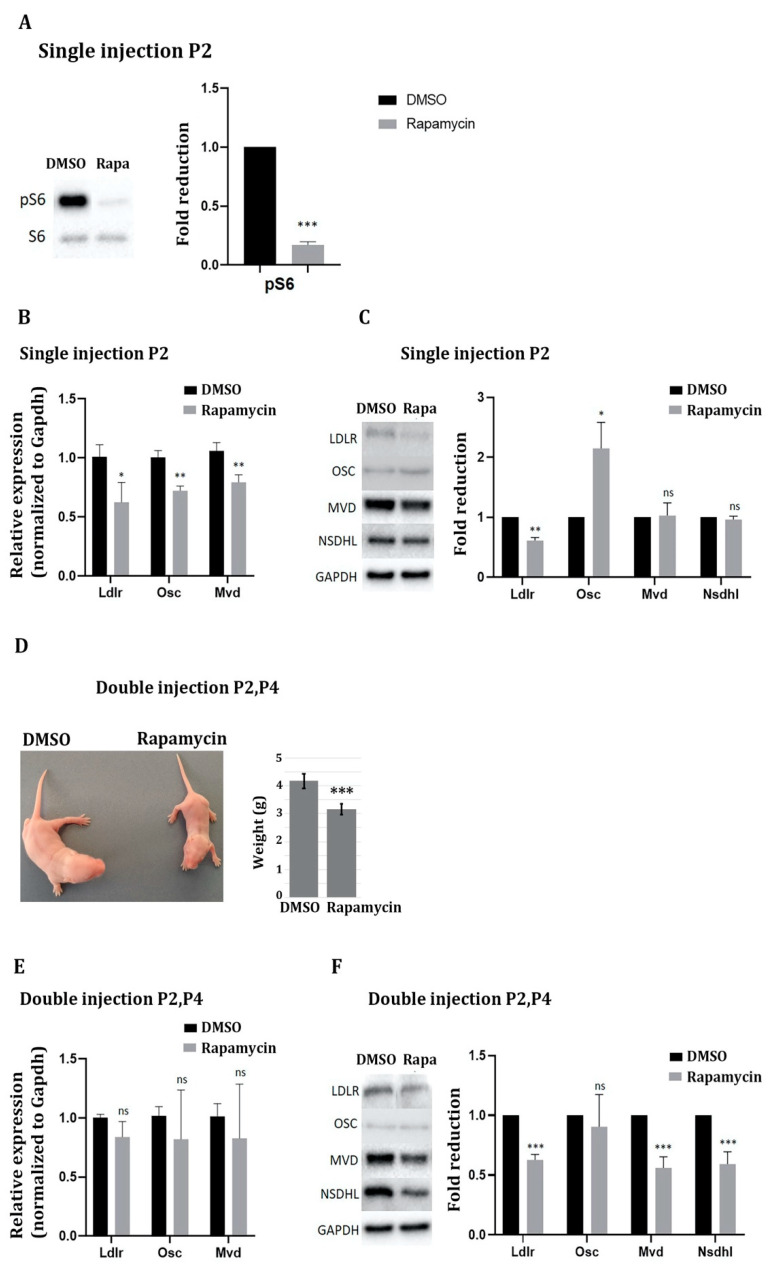
mTORC1 inhibition in the postnatal cerebral cortex downregulates cholesterol biosynthesis genes in vivo. Mice at P2 received either a single injection or received a total of two injections of DMSO or rapamycin at a dose of 1 mg/kg every two days (P2 and P4). Twenty-four hours after the single injection and 48 h after the double injection, total RNA and protein were extracted from the cerebral cortex and RT-qPCR and Western blot experiments were performed. (**A**) Inhibition of mTORC1 activity was confirmed by Western blot analysis of the mTORC1 downstream effector pS6 (compared to S6) 24 h after a single rapamycin injection. The DMSO and rapamycin treated samples were not loaded on adjacent lanes but were derived from the same gel. (**B**,**E**) RT-qPCR to detect RNA expression of the three genes *Ldlr*, *Osc*, and *Mvd* after single (**B**) or double (**E**) i. p. injections of rapamycin compared to DMSO; mRNA expression was normalized to Gapdh. (**C**,**F**) Immunoblot analysis to detect Ldlr, Osc, Mvd, and Nsdhl protein expression after one (**C**) or two (**F**) i. p. injections of rapamycin compared to DMSO and normalized to GAPDH. (**D**) Mice that had received a double injection of rapamycin at P2 and P4 were smaller than control mice at P6 and weighed less. Data represent the average of six biological replicates. For statistical analyses, Student’s *t*-test was used. * *p* < 0.05; ** *p* < 0.01; *** *p* < 0.001.

**Figure 5 ijms-22-06034-f005:**
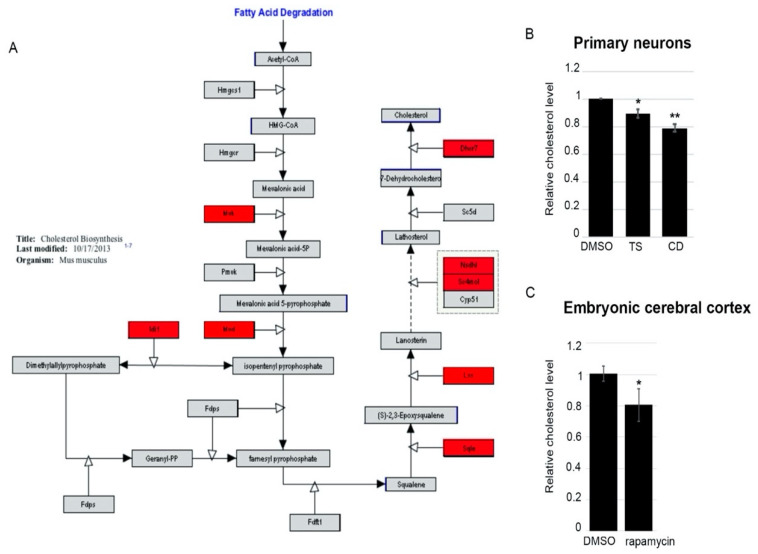
mTOR inhibition reduces cholesterol levels in primary neurons and the developing cerebral cortex. (**A**) Schematic overview of the cholesterol synthesis pathway in the mouse. Genes identified as being downregulated in the RNA-Seq experiment after 5 h of temsirolimus treatment are marked in red. (**B**) Changes in cholesterol levels were identified in DIV 6 primary cortical neurons treated with DMSO, temsirolimus (both for 48 h), or cyclodextrin (for one hour) using an Amplex Red Cholesterol Assay Kit. TS, temsirolimus; CD, cyclodextrin. (**C**) Changes in cholesterol levels after i.p. injection of 1 mg/kg rapamycin or DMSO on three consecutive days (E14.5–E16.5). * *p* < 0.05; ** *p* < 0.01.

**Figure 6 ijms-22-06034-f006:**
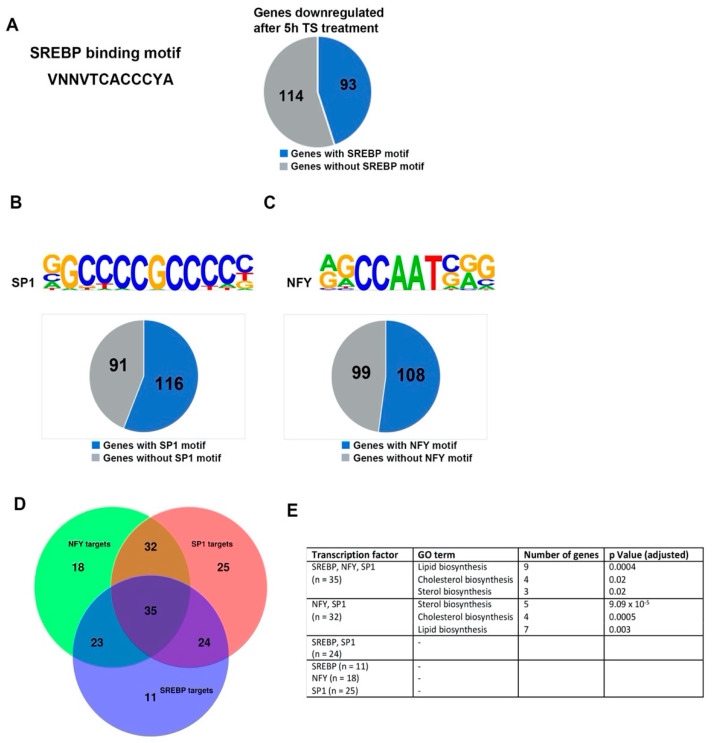
Analysis of transcription factor binding in mTORC1 inhibited neurons. (**A**) Pie chart of the genes downregulated after 5 h temsirolimus treatment, which contain SREBP binding sites in their promoters. For SREBP binding site identification, the software findM was used. (**B**,**C**) Pie charts of the genes which were downregulated after 5 h temsirolimus treatment and contained SP1 (**B**) or NF-Y (**C**) binding sites in their promoters. The software HOMER v4.9 was used to screen the promoters of the downregulated genes for the presence of transcription factor binding sites. (**D**) Venn diagram depicting downregulated genes that share binding sites for two or three of the transcription factors SREBP, SP1, and NF-Y in their promoters. (**E**) GO term analysis of the genes which contain different combinations of transcription factor binding sites in their promoters.

## Data Availability

The FASTQ files from the RNA-Seq and ATAC-Seq data were deposited in the NCBI Sequence Read Archive (SRA) under the BioProject accession number PRJNA719973.

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
