# Peer review of "mTOR Driven Gene Transcription Is Required for Cholesterol Production in Neurons of the Developing Cerebral Cortex"

_ijms, 2021, doi:10.3390/ijms22116034_

Round 1

Reviewer 1 Report

The mTORC1 signaling complex has essential roles in controlling cell growth, proliferation, and metabolism. Dysregulated mTORC1 signaling specifically in the brain causes a variety of neurodevelopmental disorders, yet how disrupted mTORC1 activity results in these developmental deficiencies remains poorly understood. In this manuscript the authors explore the role mTORC1 signaling has in regulating expression of the genes necessary for the synthesis of sterols and cholesterol in neurons and brain tissue. While mTORC1 activates a number of transcription factors, including the SREBPs, to regulate sterol/cholesterol synthesis in other tissues, how this occurs in the brain remains unknown. The authors demonstrate that mTORC1 inhibition in cortical neurons with the selective inhibitor temsirolimus results in selective downregulation of sterol/cholesterol biosynthesis genes, but not other metabolic genes linked to mTORC1 signaling, including glycolytic genes. The authors then go on to perform in vivo administration of mTORC1 inhibitor to demonstrate that these same sterol/cholesterol genes are selectively downregulated, strongly suggesting that mTORC1 signaling is essential for normal sterol/cholesterol synthesis during development. They then identify key consensus transcription factor binding sites for transcription factors that are known to play a role in this process, including SREBP, as well as the additional transcription factors NFY and SP1.

            I found the work presented in this manuscript important to the field. I believe after careful revision this manuscript would be acceptable for publication in International Journal of Molecular Sciences. The specific areas of concern are outlined below.

Scientific concerns

  1. On page 3, the authors jump to the assumption that the gene expression changes they are monitoring are due to alterations in transcription. This is a hasty and precocious assumption since these mRNA expression differences could just as easily be explained by changes in the rate of mRNA decay. Later in the manuscript, the authors address this issue by monitoring mRNA decay, and they incorporate this data into the Supplemental file. I would highly suggest that they insert this information at this point in the manuscript so that demonstrate early, right at the point where they discuss differential transcription as the sole reason for the changing gene expression, that the results are indeed due to altered transcription. An additional chromatin immunoprecipitation (ChIP) for RNA Polymerase II at the promoters of the 3-4 core genes they are monitoring also would strengthen their argument considerably.
  2. The authors should include a positive control (beyond S6ph) for the Torin 1 treatment. A rapamycin-resistant mTORC1 substrate such as 4EBP would be acceptable, but the reader needs to see that both the rapamycin sensitive and insensitive mTORC1 substrates are being inhibited with Torin 1.
  3. For Figure 2B, labeling of the y-axis would be very helpful in interpreting the data. At first glance, it is not immediately apparent what the authors are plotting here.
  4. The immunoblot in Figure 4A looks like it has been split, ie the authors have merged two separate immunoblots into one figure (there is an almost imperceptible line between the DMSO and Rapa panels). If this is indeed the case, the authors need to clearly indicate this to be so.

Editorial concerns

  1. This manuscript could use some editorial revisions throughout. There are sentences throughout the manuscript where commas should be placed and they are not, and there is some issues with sentence structure. One notable example is on page 2, line 53 “In this context…..”. This sentence is difficult to process, and it reads as one long run-on sentence.
  2. The sentence on page 2, line 69 should be integrated into the preceding paragraph as it does not fit well as a standalone sentence.
  3. On page 3, line 88, “…. 0.5 log2foldChange…”. This should be corrected.

Author Response

Dear Reviewer,

Thanks a lot for your comments and constructive criticism. Below you can find our point-by-point response.

Scientific concerns

  1. On page 3, the authors jump to the assumption that the gene expression changes they are monitoring are due to alterations in transcription. This is a hasty and precocious assumption since these mRNA expression differences could just as easily be explained by changes in the rate of mRNA decay. Later in the manuscript, the authors address this issue by monitoring mRNA decay, and they incorporate this data into the Supplemental file. I would highly suggest that they insert this information at this point in the manuscript so that demonstrate early, right at the point where they discuss differential transcription as the sole reason for the changing gene expression, that the results are indeed due to altered transcription. An additional chromatin immunoprecipitation (ChIP) for RNA Polymerase II at the promoters of the 3-4 core genes they are monitoring also would strengthen their argument considerably.

We agree with the reviewer. We have now included the mRNA decay analysis on page 3. We agree with the reviewer that ChIP experiments would be a good strategy to investigate the mechanisms of mTOR mediated transcriptional regulation in more detail, and we will certainly do this in follow-up studies. However, since ChIP experiments are not trivial and take time to establish this would go beyond the scope of this manuscript.

  1. The authors should include a positive control (beyond S6ph) for the Torin 1 treatment. A rapamycin-resistant mTORC1 substrate such as 4EBP would be acceptable, but the reader needs to see that both the rapamycin sensitive and insensitive mTORC1 substrates are being inhibited with Torin 1.

We have now included both pS6 and p4EBP in Supplemental figure S2A.

  1. For Figure 2B, labeling of the y-axis would be very helpful in interpreting the data. At first glance, it is not immediately apparent what the authors are plotting here.

We have now labeled the y-axis.

  1. The immunoblot in Figure 4A looks like it has been split, ie the authors have merged two separate immunoblots into one figure (there is an almost imperceptible line between the DMSO and Rapa panels). If this is indeed the case, the authors need to clearly indicate this to be so.

The samples were not loaded on adjacent lanes but were derived from the same gel. We have now indicated this in the legend to figure 4A.

Editorial concerns

  1. This manuscript could use some editorial revisions throughout. There are sentences throughout the manuscript where commas should be placed and they are not, and there is some issues with sentence structure. One notable example is on page 2, line 53 “In this context…..”. This sentence is difficult to process, and it reads as one long run-on sentence.

We revised the manuscript and placed commas where appropriate. We have now changed the sentence on page 2, line 53.  

  1. The sentence on page 2, line 69 should be integrated into the preceding paragraph as it does not fit well as a standalone sentence.

We have now integrated this sentence into the preceding paragraph.

  1. On page 3, line 88, “…. 0.5 log2foldChange…”. This should be corrected.

We have now corrected this.

Reviewer 2 Report

The studies by Schule et al examine the role of mTOR in neurons in vitro and in the developing cerebral cortex in vivo.  Using rapamycin at two time points (5 and 24 hr) they compared gene expression changes and found that the expression of genes involved in cholesterol metabolism are diminished.  In contrast, they found no effect on glycolysis genes unless Torin, which inhibits both mTORC1 and mTORC2, was used.  In vivo injections also revealed interesting results on reduction of cholesterol metabolism related genes, although protein levels were not significantly changed.  However, when treated for three consecutive days, although there was no change in gene expression, protein expression of these genes diminished.  Prolonged in vivo treatment also led to smaller mice and decreased protein expression of cholesterol biosynthesis-related genes.  Next, they also measured cholesterol production in vitro and in vivo upon rapamycin treatment and found small but significant reduction.  Finally, they examined how mTOR inhibition could lead to gene expression changes in the cholesterol-related genes and found using in silico analysis that a majority of genes that have altered expression contain binding sites for the transcription factors SREBP,  SP1 and NF-Y.  Altogether they conclude that mTOR  regulates transcription of genes involved in cholesterol biosynthesis in neurons and developing brain.

The studies are interesting and well presented. 

 Can the authors comment in the Discussion why protein levels change only after more prolonged inhibition. 

Figure 3 C and E: Indicate that these figures are measuring protein levels (as opposed to mRNA levels in the other panels.

Author Response

Dear Reviewer,

Thanks a lot for your comments and constructive criticism. Below you can find our point-by-point response.

Can the authors comment in the Discussion why protein levels change only after more prolonged inhibition.

This is now included on page 14, lane 403. 

Figure 3 C and E: Indicate that these figures are measuring protein levels (as opposed to mRNA levels in the other panels.

We have now indicated that mRNA and protein levels were measured in the respective figures.

Reviewer 3 Report

The manuscript entitled 'mTOR driven gene transcription is required for cholesterol production in neurons of the developing cerebral cortex' by Schuele et al. investigates mTOR-dependent gene transcription in developing neurons with regard to cholesterol balance and control. With the help of different methodological approaches (among which are RNA-sequencing, RT-PCR, Western Blotting and cholestrol assay), Schuele et al. show that mTORC1 inhibition by temsirolimus and/or rapamycin in cultured neurons and/or mouse embryos/postnatal pups causes a significant dysregulation of many genes among which were several genes of the cholesterol synthesis but not glycolysis pathway. Specifically, the authors then focused on mTORC1-dependent inhibition of Ldlr, Osc, Mvd, Dhcr7 or Nsdhl on either mRNA or protein level. Finally, the authors show that mTORC1 inhibition is accompanied by a reduction of cholesterol levels in cultured neurons and in the embryonic mouse brain.

While the study is appropriately designed and the introduction is written in a clear and concise way there are several minor concerns and experimental inconsistencies regarding the manuscript:

1) Why did the authors choose to analyze Dhcr7 on mRNA and Nsdhl on protein level (related to Figure 2)?

2) Why did the authors omit Dhcr7 analysis after a single rapamycin injection at P2 while having shown rapamycin-dependent differences in Dhcr7 levels in embryonic brains? If the data were not collected the lack of these data must at least be discussed.

3) A single rapamycin injection at E16.5 causes no change in Osc protein levels whereas the same treatment at P2 causes a significant increase in Osc protein levels. Theses differences need to be addressed and discussed.

4) The authors compared single rapamycin injections to double or triple rapamycin injections and argue that high protein stability could be the reason for observing reduced Ldlr, Osc etc levels on mRNA but not on protein level. Why did the authors choose a triple rapamycin injection (three consecutive days) in embryonic and a double rapamycin injection (at P2 and P4, kill at P6) in postnatal pups? These differences in the experimental design need to be discussed.

5) Have the authors also tested cholesterol levels in postnatal pups after rapamycin treatment?

6) Can the authors explain why the number of open chromatin regions detected by ATAC-seq is twofold higher in the second replicate compared to the first replicate?

7) Have the authors tested for a normal distribution of their data points before choosing a t-test for statistical analysis? This needs to be indicated.

8) Spaces, upper/lower cases and consistent use of abbreviations must be checked.

Author Response

Dear Reviewer,

Thanks a lot for your comments and constructive criticism. Please find our point-by-point response below.

1) Why did the authors choose to analyze Dhcr7 on mRNA and Nsdhl on protein level (related to Figure 2)?

In these initial experiments we did not have good working antibodies detecting DHCR7 and NSDHL.

2) Why did the authors omit Dhcr7 analysis after a single rapamycin injection at P2 while having shown rapamycin-dependent differences in Dhcr7 levels in embryonic brains? If the data were not collected the lack of these data must at least be discussed.

Due to limited material at P2 and the fact that we did not have a working Dhcr7 antibody and could, therefore, not test DHCR7 protein expression, we decided to omit analysis of Dhcr7 mRNA expression at P2.

3) A single rapamycin injection at E16.5 causes no change in Osc protein levels whereas the same treatment at P2 causes a significant increase in Osc protein levels. Theses differences need to be addressed and discussed.

We have now addressed this in the new version of the manuscript on page 14 lane 405.

4) The authors compared single rapamycin injections to double or triple rapamycin injections and argue that high protein stability could be the reason for observing reduced Ldlr, Osc etc levels on mRNA but not on protein level. Why did the authors choose a triple rapamycin injection (three consecutive days) in embryonic and a double rapamycin injection (at P2 and P4, kill at P6) in postnatal pups? These differences in the experimental design need to be discussed.

In the literature it was already shown that at postnatal stages efficient mTOR inhibition could even be achieved by rapamycin injections every other day. To spare the mice we decided to inject rapamycin every other day.

5) Have the authors also tested cholesterol levels in postnatal pups after rapamycin treatment?

We have not done this so far but will perform deeper investigations on changes in cholesterol levels after mTOR inhibition in a follow-up study.

6) Can the authors explain why the number of open chromatin regions detected by ATAC-seq is twofold higher in the second replicate compared to the first replicate?

The transposition reaction was performed in two different biological replicates (each biological replicate comprised of temsirolimus and DMSO treated cells). The efficiency of transposition reaction could vary between batches particularly if they are treated on distinct days. This technical difference could be the main reason for difference in number of open accessible regions detected. However, most importantly we compared temsirolilmus versus DMSO treated cells per batch and we observed relatively similar amount of accessible regions between the two conditions. When we compared the association of each condition (Ts and DMSO) with the cholesterol genes of interest we did not find significant changes in both of the biological replicates.

7) Have the authors tested for a normal distribution of their data points before choosing a t-test for statistical analysis? This needs to be indicated.

For small sample sizes, like in our study, testing for a normal distribution of the data is not recommended, due to the low power. We have, therefore, not tested for a normal distribution.

8) Spaces, upper/lower cases and consistent use of abbreviations must be checked.

We have now checked the manuscript for consistency.